# The influence of the baseline drift on the resulting extinction values of a CAPS PMex

Sascha Pfeifer[1], Thomas Müller[1], Andrew Freedman[2], and Alfred Wiedensohler[1]

[1]Leibniz Institute for Tropospheric Research, Permoserstraße 15, 04318 Leipzig, Germany
[2]Aerodyne Research Inc., 45 Manning Road, Billerica, MA 01821-3976, USA

**Correspondence:** S. Pfeifer (pfeifer@tropos.de)

**Abstract.** The effect of the baseline drift on the resulting extinction values of three CAPS PMex monitors with different wavelengths and the respective correlation with $NO_2$ was analysed for an urban background station. A drift of more than $0.8\,\mathrm{Mm}^{-1}\,\mathrm{min}^{-1}$ was observed for ambient air, with high probability caused by traffic emissions driven changes in carrier gas composition.

The baseline drift leads to characteristic measurement artefacts for particle extinction. Artificial particle extinction values of approximately $4\,\mathrm{Mm}^{-1}$ were observed using a baseline period of $5\,\mathrm{min}$. These values can be even higher for longer baseline periods.

Two methods are shown to minimize this effect. Modified continuous baseline values are calculated in a post-processing step using simple linear interpolation and cubic smoothing splines. Both methods are useful to reduce artefacts, although the use of cubic smoothing splines gives slightly better results. The extinction artefacts are diminished and the effective scattering of the resulting extinction values is reduced by about $50\,\%$.

## 1 Introduction

Aerosol particles affect the global albedo or radiation balance of the earth by interacting with solar and thermal radiation through absorption and scattering processes. In order to estimate the influence on the climate, it is therefore important to determine the optical properties of the atmospheric aerosol with sufficient accuracy. In particular, the aerosol scattering $\sigma_{sp}$, absorption $\sigma_{ap}$ and extinction $\sigma_{ep}$ coefficients, from which the single scattering albedo $\omega = \sigma_{sp}/\sigma_{ep}$ is derived, are important parameters.

Various in-situ measurement techniques exist for the respective parameters. In the past, cavity ring-down technology was used, to measure the $\sigma_{ep}$ directly (Brown, 2003). A very similar measurement method is cavity attenuated phase shift (CAPS) technique. A square wave modulated light of a light emitting diode (LED) is injected in an optical cavity, defined by two high reflectivity mirrors ($R > 0.999$) in a distance of 26 cm. The phase shift of the distorted signal caused by the effective optical path is measured by a vacuum photo diode on the opposite side. This is a robust, state-of-the-art and commercially available measurement method, which is also used as gas monitor to measure ambient $NO_2$ concentration (Kebabian et al., 2005, 2008).

The CAPS PMex (Massoli et al., 2010) enables the measurement of the $\sigma_{ep}$ by periodically changing between ambient air (normal measuring period) and particle-free air (baseline period). The Rayleigh scattering value for air $\sigma_{ea}$ at given temperature and pressure condition is subtracted from the respective raw signals, called total loss ($loss$). The resulting values are averaged over the period of baseline duration. This value is called last baseline ($lastbaseline$), which not only depends on device parameters, in particular the degree of contamination of the cavity mirrors, but also on the concentration of absorbing gases.

The resulting values for particle extinction $\sigma_{ep}$ for the following normal measuring period is calculated as:

$$\sigma_{ep}(T,P) = \frac{loss(T,P) - \sigma_{ea}(T,p) - lastbaseline}{g}, \tag{1}$$

where $g$ is the geometry factor, considering the effect of the purge air on the effective optical path length. The crucial point is that the measurement is calculated using the baseline values, which are assumed to be constant for a certain period and lag behind in time.

A detailed description of the Instrument is given by Massoli et al. (2010). CAPS PMex has already been compared and characterized in combination with other instruments (Petzold et al., 2013) and used in various campaigns (Yu et al., 2013; Perim de Faria et al., 2017).

Although the instrument delivers a satisfying performance, Massoli et al. (2010) already mentioned deviations due to a baseline drifts. Motivated by this aspect, the aim of this work is to examine the effect of the baseline drift in more detail. For this purpose, exemplary measurements at an urban background station are analysed. In addition, a possible approach in post-processing is proposed to reduce the influence of the baseline drift.

## 2 Experimental set-up

In order to analyse the influence of the baseline drift on the resulting extinction values, measurements of ambient air were carried out at the Leibniz Institute for Tropospheric Research (Leipzig, Germany) over a period of two weeks. The measurement site, classified as an urban background station, is influenced by two main roads and rail traffic, as well as a small gas power plant.

The measurements were performed with three different CAPS PMex monitors of different wavelengths: CAPS-blue ($450\,\mathrm{nm}$), CAPS-green ($530\,\mathrm{nm}$) and CAPS-red ($630\,\mathrm{nm}$). The sampling rate for all CAPS PMex was set to $1\,\mathrm{Hz}$. The baseline period was set to $5\,\mathrm{min}$ with $60\,\mathrm{s}$ duration and $30\,\mathrm{s}$ flushing time.

In addition, the concentration of equivalent black carbon (eBC) was measured with a multi-angle absorption photometer (MAAP) at the same inlet system with a time resolution of $1\,\mathrm{min}$. Furthermore, the NOx concentration was measured with an APNA-370 Ambient NOx Monitor at an separate inlet at the roof top with $3\,\mathrm{min}$ resolution.

To analyse the influence of the variability of the gas concentration of the carrier gas and to rule out the influence of aerosols on the resulting extinction values, an additional filter was installed upstream of the three CAPS PMex. According to a zero filter test, values are expected to be around zero for the whole period. Deviations from this indicate a systematic error.

Before and after measurements the quality of the CAPS PMex were checked by a comparison with a thoroughly and regularly calibrated reference nephelometer (Ecotech Aurora 4000) using $CO_2$ as high span gas. For this purpose non-absorbing ammonium sulfate particles were used. The truncation error in the nephelometer has been corrected using the method of Müller et al. (2011). The values were adjusted to the corresponding wavelength of the CAPS PMex (450, 530 and 630 nm) by using scattering Angström exponent. Nevertheless relatively small particles were generated (mean size of approx. 50 nm) to minimize the effect of truncation. Analogous to the comparison of the measured and Mie calculated theoretical values using mono-disperse particles and a reference CPC (Petzold et al., 2013), correction factors can be derived by comparing the truncation-corrected scatter values of the reference nephelometer with the respective measured extinction values. The factors represent a correction of internal calibration, which primarily consider the influence of the variable ratio of purge air and sample air flow rate. In order to reduce the influence of potential non-linear effects of CAPS PMex, only values less than 500 were used for this analysis.

## 3 Results

### 3.1 Variability of background signal

Time series for the loss signals of all three CAPS PMex, as well as the eBC and $NO_2$ concentrations are shown in Fig. 1. The CAPS-blue shows a significant variability of the loss signal, with background values of $585\,\mathrm{Mm}^{-1}$ and peaks up to $635\,\mathrm{Mm}^{-1}$. The CAPS-green shows an identical behaviour but with a lower amplitude, with background values of $380\,\mathrm{Mm}^{-1}$ and peaks up to $400\,\mathrm{Mm}^{-1}$. The values for the CAPS-red are independent and rather stable ranging from $480\,\mathrm{Mm}^{-1}$ up to $484\,\mathrm{Mm}^{-1}$. During the two week period the maximum eBC and $NO_2$ concentration where $5\,\mathrm{\mu g\,m}^{-3}$ and $45\,\mathrm{ppb}$, respectively.

In table 1 the corresponding correlations coefficients for the time series are shown. As already expected from Fig. 1, both loss values of CAPS-blue and CAPS-green are highly correlated ($R^2 = 0.845$). The highest correlation is found between the loss of CAPS-blue and the $NO_2$ concentration ($R^2 = 0.945$), while the correlation of CAPS-blue with eBC is $R^2 = 0.785$. The values of loss from CAPS-red was found to be uncorrelated to the other variables. On average, the time series for loss (CAPS-blue and CAPS-green) as well as eBC and $NO_2$ show increased values in the night with a maximum in the late evening and another maximum in the morning. A minimum occurs at noon. This behaviour is repeated every day, with the exception of the weekend (21-22 September). In general, these values follow the daily pattern, resembling traffic rush hours and development of planetary boundary layer. Because of the total filter upstream of the CAPS PMex the measured variability of loss signal is not due aerosol particles. This variability can only be explained by changes of the ambient air, which is likely based on changes of $NO_2$ concentration due to traffic related emission. The steady increase of the loss of CAPS-green in the second week is significant. The reason for this is unknown. Because a particle filter was used, it can be excluded that this is based on contamination by aerosol particles on the cavity mirrors.

However, the variability of loss signal can be quite high, whereby the ascending flank is steeper than the descending flank. For the CAPS-blue the rate of change was in the range of $-0.72\,\mathrm{Mm}^{-1}\,\mathrm{min}^{-1}$ to $0.83\,\mathrm{Mm}^{-1}\,\mathrm{min}^{-1}$ (99% percentile). The values for maximum rate of change were $-1.78\,\mathrm{Mm}^{-1}\,\mathrm{min}^{-1}$ and $4.15\,\mathrm{Mm}^{-1}\,\mathrm{min}^{-1}$ respectively. The influence on CAPS-green is lower but still noticeable with values in the range of $-0.18\,\mathrm{Mm}^{-1}\,\mathrm{min}^{-1}$ to $0.22\,\mathrm{Mm}^{-1}\,\mathrm{min}^{-1}$ (99% percentile).

Before and after the measured time series the comparison of CAPS PMex and reference nephelometer show a small but very stable deviation, exemplary shown in Fig. 2. The devices show slightly too high values in the range of 3–4 %, 6–8 %, and 6–7 % for the blue, green and red, respectively.

## 3.2  Artefacts from internal baseline correction

As previously mentioned, variations in the baseline by rapid change in the concentration of absorbing gases may occure with values up to $4\,\mathrm{Mm}^{-1}\,\mathrm{min}^{-1}$. Hence, the assumption of a constant baseline value for internal data processing may cause uncertainties.

Any changes of the baseline during a normal measuring period due to changes in gas composition are immediately misinterpreted as aerosol extinction. Furthermore, due to the forward extrapolation, the internal $lastbaseline$ value is phase shifted to the supposedly correct value.

Figure 3 shows a one-hour excerpt from the time series of CAPS-blue. A smooth and continuous increase of the loss signal from $590\,\mathrm{Mm}^{-1}$ to $620\,\mathrm{Mm}^{-1}$ for the measuring period is observed. The time series for the $lastbaseline$ value shows a step-like function, which is phase-shifted relative to the loss signal. This results in artificial extinction values of up to $5\,\mathrm{Mm}^{-1}$ with a saw-tooth structure. For a continuously increasing loss signal the extinction values are strictly positive. The opposite is true for decreasing loss signals. Due to the stronger increase than decrease for loss signal, the resulting extinction values are not symmetrically distributed.

It is possible to reduce these artefacts by using interpolation methods. Two different procedures were considered. The first one is a simple and often used linear interpolation method. A second, and potentially better, alternative is the use of cubic smoothing splines. For this post-processing the loss values for the baseline period were extracted, subtracted by the corresponding Rayleigh value and used as predictor variables for interpolation with cubic smoothing splines. The cubic smoothing spline function (smooth.spline) provided by R (R Core Team, 2013) was used for this purpose. A free smoothing parameter ($spar$) must be chosen, which depends on many factors, e.g. baseline period and duration but also on sampling rate and device noise . Therefore, a suitable parameter must be found for each individual device and application. For the case with $1\,\mathrm{Hz}$ sampling rate, a baseline period of $5\,\mathrm{min}$ and a duration of $1\,\mathrm{min}$, the smoothing parameter used were 1.1, 1.3, and 1.4 for the blue, green and red, respectively. These values were determined by minimizing the artefacts of a separate test dataset. Alternatively it is also possible to automatically determine a suitable smoothing parameter from the time series of baseline using for example the implemented generalized cross-validation method (GCV). The resulting values of the automatically calculated smoothing parameters using the GCV method do not differ significantly from the first method with values of 1.06 (blue), 1.25 (green) and 1.30 (red). Furthermore, all distinct data points with $1\,\mathrm{Hz}$ sampling rate were used (all.knots=TRUE). All other parameters were set to default. A complete description of the function can be found in the R Documentation (R Core Team, 2013).

It should be emphasized that the use of any interpolation method to recalculate the baseline has its limits. Only trends that can be estimated from the baseline data can be reproduced for $lastbaseline$. It is impossible to reproduce any faster fluctuations that are not covered by the selected baseline period and duration. Furthermore, when using the cubic smoothing splines there

is the possibility that under extreme conditions with strongly fluctuating baseline trends the method can lead to erroneous overshot structures. In these cases, the first step should be the readjustment the baseline settings.

If the requirements are fulfilled, these approaches result in a continuous time series of current baseline values, without phase-shift relative to the loss signal (see Fig. 3). As expected, the result is slightly better when cubic smoothing splines are used, as this is a continuously differentiable function. Another important difference is that with cubic smoothing splines trends during a baseline measurement are considered and are therefore reproducible. In contrast to this, the linear interpolation method uses only one average value per baseline measurement, analogous to the internal procedure. As a result, there are individual cases where the linear interpolation does not lead to any improvement of the extinction values, but there are also cases where the improvement corresponds to that of the cubic smoothing spline. However, in both case the resulting extinction values improve significantly. In Fig. 4 the resulting histograms and statistical parameters for particle extinction for all instruments and the entire time series are shown. As expected, the mean value remains almost unchanged at values close to zero. But the distribution becomes narrower and more symmetrical. For the CAPS-blue the standard deviation is reduced by $43\%$ using the linear interpolation and $50\%$ using cubic smoothing splines. For CAPS-green the reduction is $19\%$ with both methods. The skewness for CAPS-blue is reduced from $2.909$ to a value of $0.756$ using the linear interpolation and $0.104$ for cubic smoothing splines. The results for the CAPS-red remain almost unchanged.

Figure 5 shows the data for the whole measurement period plotted as secondary Allan standard deviation values versus integration time using uncorrected and corrected data for all three wavelengths. Allan plots show the effective noise levels as a function of integration time and allow one to separate the effects of baseline drift from short term noise. Typically, data for these plots are taken without baseline periods in order to gauge the effects of baseline drift. In the plot shown here, the data has been corrected for drifting baselines and thus provides a demonstration of how well baseline subtraction actually works. In the case of CAPS-red where the effects of $NO_2$ are minimal, there is little difference between the results with and without post-processing. However, for CAPS-blue und CAPS-green, the improvement is substantial. At $450\,nm$, where $NO_2$ absorption is maximized, and to a lesser extent at $530\,nm$, without correction, measurement precision is completely limited by the intervals between baseline measurements at a level far above short term noise levels. However, with the correction scheme, the data can be integrated for long periods of time in order to improve precision. For instance, at $450\,nm$, the precision is improved by a factor of approximately 2.8 using the linear interpolation method and 3.7 using cubic smoothing splines.

## 4 Conclusions

The effect of the baseline drift on the measurement values of three different CAPS PMex for an urban background station was analysed. The drift can be up to $0.8\,Mm^{-1}\,min^{-1}$. For internal data processing, it is assumed that the baseline does not change for the following measurement period. In combination with a fast variable background signal or baseline drift, this can lead to measurement artefacts. The effect of baseline drift is additive, therefore, the relative error is higher for low particle extinction.

The use of linear interpolation or cubic smoothing splines to calculate the current baseline values are more adequate methods for a variable background. Both procedures lead to improved values, although the result for cubic smoothing splines is slightly

better. Artefacts for particle extinction almost disappear and variability decreases. Any other approach that provides a continuous time series for the baseline without phase shift seems just as useful. The use of interpolation methods is a general approach for instruments which are affected by a drift, but due to the measuring principle of the CAPS PMex this fact is especially important.

If the change of the background signal is relatively slow, these methods allows to reduce the frequency of baseline periods and thus reduce number of position changes of the built-in ball valve extending its lifetime. On the other hand, in the majority of cases, the background variability is unknown and the ambient aerosol and the composition of the carrier gases may be closely coupled (e.g. near traffic emission). From this it follows that the measuring and baseline period should be equally weighted, if one considers the background signal as equivalent. The use of a gas monitor in parallel operation can serve as a reference to adjust the baseline period. However, these interpolation methods, in particular cubic smoothing splines, can be used to take into account the continuous change of the background signal and improve the quality of the resulting extinction values.

*Acknowledgements.* This work was carried out in the context of the 16ENV02 Black Carbon project of the European Union through the European Metrology Programme for Innovation and Research (EMPIR). This project has received funding from the EMPIR programme co-financed by the Participating States and from the European Union's Horizon 2020 research and innovation programme.

**Table 1.** Correlation coefficients of the loss values of the three CAPSs (450, 530 and 630 nm), and the eBC and $NO_2$ concentrations.

|  | loss (blue) | loss (green) | loss (red) | eBC | $NO_2$ |
|---|---|---|---|---|---|
| loss (blue) | 1.000 | 0.845 | 0.255 | 0.785 | 0.945 |
| loss (green) | - | 1.000 | -0.201 | 0.605 | 0.844 |
| loss (red) | - | - | 1.000 | 0.221 | 0.175 |
| eBC | - | - | - | 1.000 | 0.773 |
| $NO_2$ | - | - | - | - | 1.000 |

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

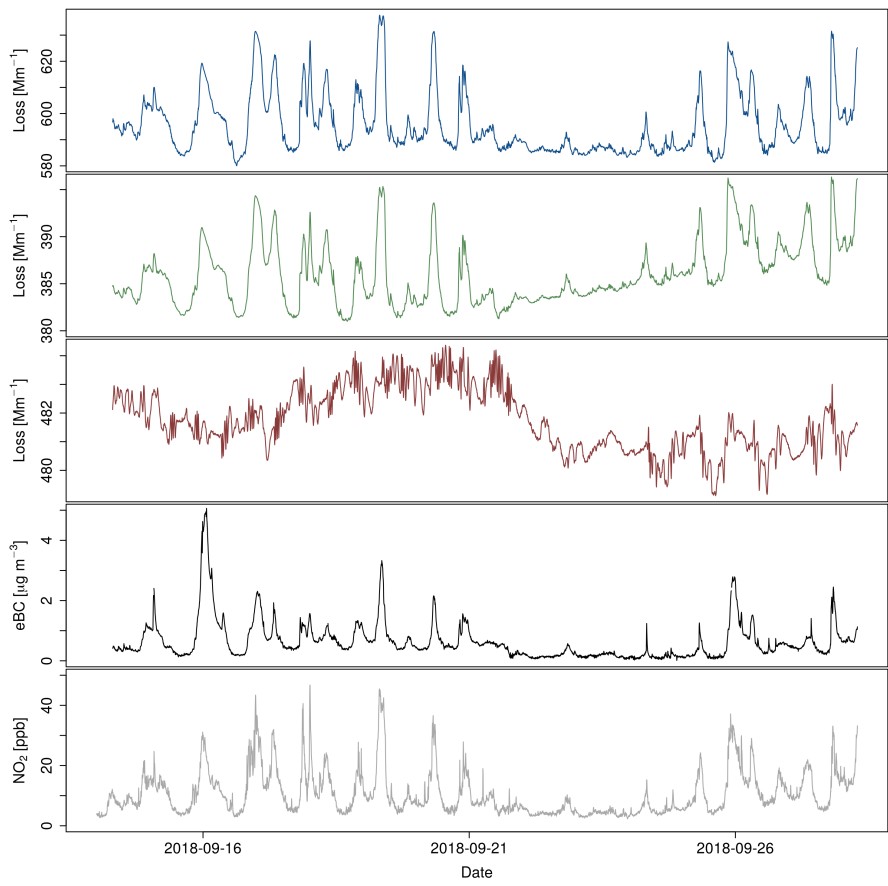

**Figure 1.** Time series of loss signal measuring particle free ambient air (450, 530 and 630 nm) and eBC and NO₂ concentration for a two week period.

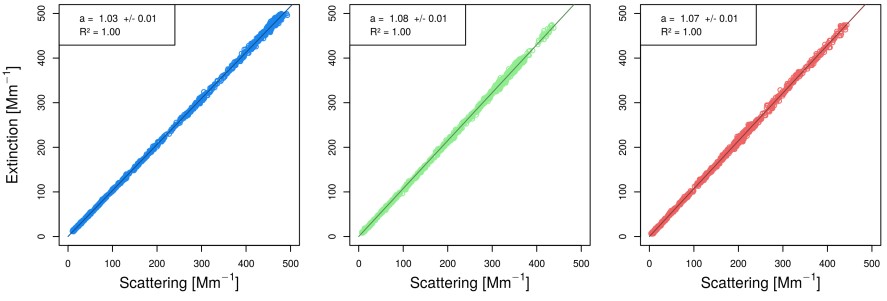

**Figure 2.** Comparison of measured extinction (450, 530 and 630 nm) and scattering measured with nephelometer (truncation corrected) for non-absorbing ammonium sulfate particles after the measured time series.

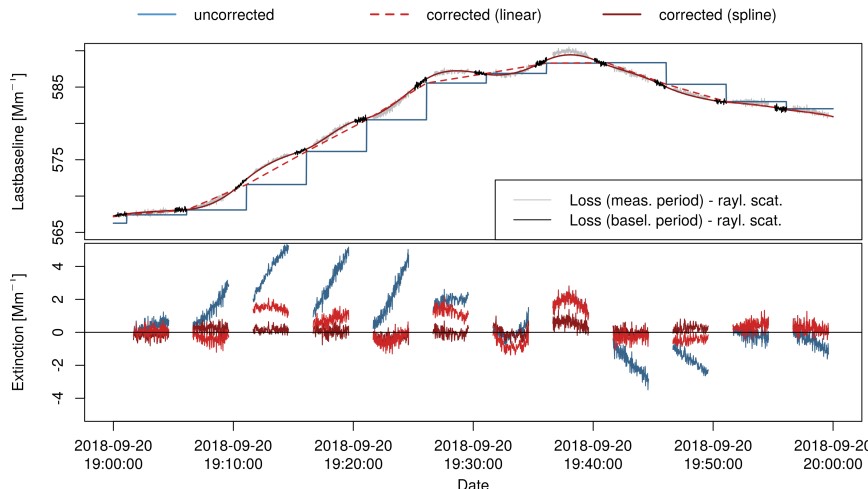

**Figure 3.** Time series of $lastbaseline$ (top) and the resulting extinction values (bottom) for the uncorrected (blue) and corrected methods, using linear interpolation (orange) and cubic smoothing splines (red), for CAPS-blue (450 nm) measuring particle free ambient air. In the top panel the loss signal subtracted by the Rayleigh scattering for the measurement (grey) and baseline period (black) are additionally shown. The resulting extinction value in the lower panel is the direct consequence of the deviation between these values the used baseline.

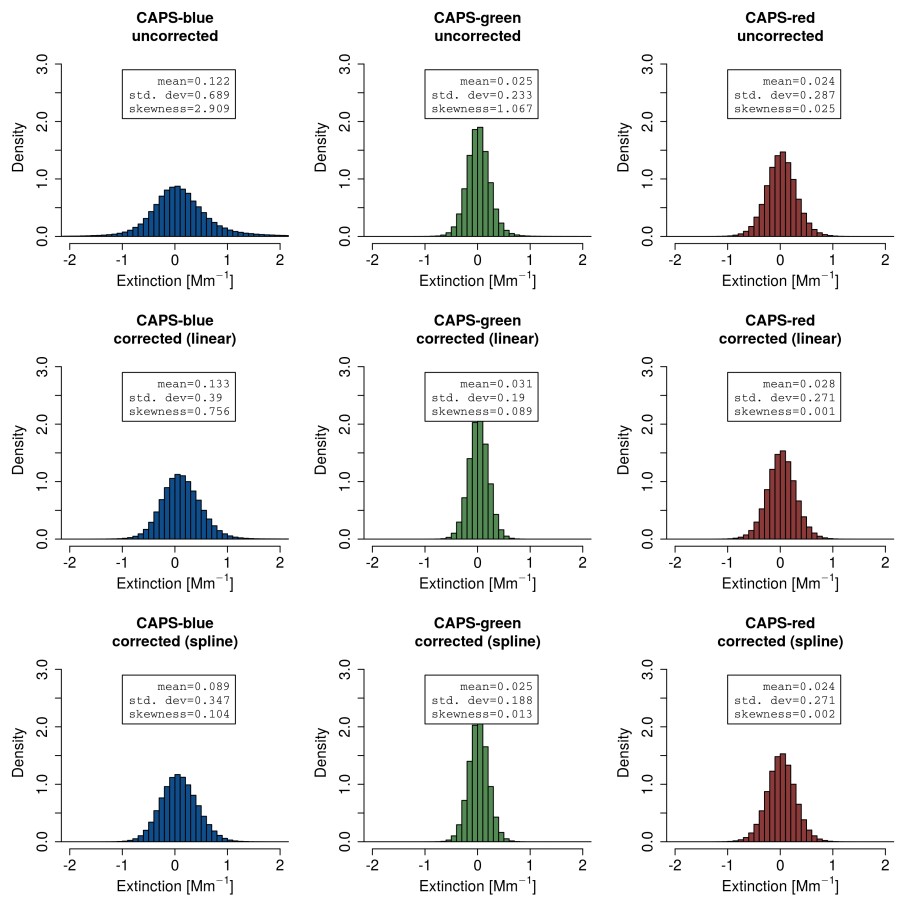

**Figure 4.** Histogram of the extinction values from measurements of particle free ambient air for all three wavelengths (450, 530 and 630 nm). The top panels show the uncorrected extinction values, while the middle and bottom panels show corrected values using linear interpolation and cubic smoothing splines, respectively. The corresponding statistical parameters of the histograms are indicated in the boxes.

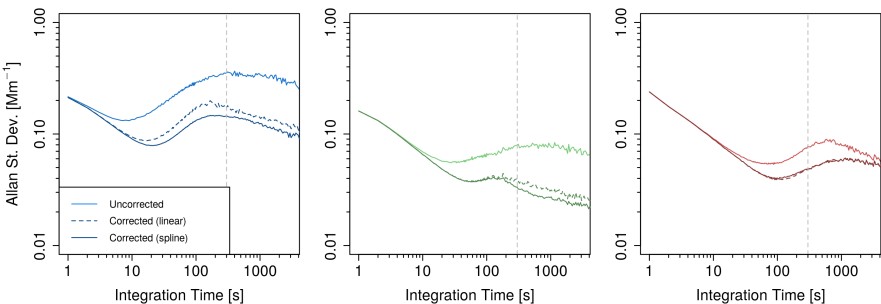

**Figure 5.** Secondary Allan standard deviation plots of extinction coefficients (450, 530 and 630 nm) for particle free ambient air. Vertical dashed lines denote time intervals of baselines (300 s).