# Peer review of "The influence of the baseline drift on the resulting extinction values of a CAPS PMex"

_Atmospheric Measurement Techniques, 2019_

## Referee Comment (RC1) · Anonymous Referee #2 · 2 Dec 2019

General comments

This paper addresses biases associated with aerosol extinction measurements from Aerodyne CAPS PMex sensors occurring in the presence of gaseous absorbers. In particular, biases become apparent when gaseous absorption varies on a timescale faster than the CAPS baseline characterisation interval. It is shown that the bias can be severe, particularly at blue spectral wavelengths where nitrogen dioxide, which is often co-located with the particulate matter of interest, absorbs strongly. The paper proposes a correction scheme that users can apply to reduce these systematic sampling biases.

It is without doubt important that users of CAPS PMex systems are aware of these biases. I would however note that these biases are in no way surprising and are already very well documented in the literature for similar research-grade instruments. Indeed,

several instruments adopting cavity ringdown-based techniques for aerosol detection have implemented specific measures to eliminate biases from gaseous absorption entirely. In my mind this does raise the question of whether this paper presents genuinely novel insight.

I find that the correction scheme proposed is at best a partial fix to the problem. It is shown to reduce error in the example data well, but there are scenarios where it is likely to provide insufficient error compensation. For example, when NO2 concentrations do not vary smoothly between baseline periods, it may be far less effective (and potentially even make things worse than the standard corrections – see below). I would have preferred the conclusion of the paper to have directed users to a more general and reliable solution, which ultimately could require physical modification of the instrument, sample conditioning (e.g. to scrub NO2/O3) or mode of operation (e.g. to run a designated monitor with a permanent filter to measure the gas phase background and variability). More focus is also needed to examine and explain the limitations of the suggested correction method. For these reasons I would recommend significant revision before the the paper is considered for publication.

Specific comments

- The suggested correction scheme works for the example data provided. However it will not work as effectively when there is significant variability in NO2 between consecutive baseline measurements. Indeed, it would appear possible that in some circumstances the applied correction could make the bias worse compared to the standard method (e.g. where the baseline increases between consecutive baseline periods but gaseous absorption decrease in the interim period). For this reason I question what applying this new correction really enables users to say with confidence about the accuracy of their resulting extinction numbers. This needs to be examined in more detail in the paper.

- To overcome the above, the authors suggest that users will need to tune their smoothing parameters based on the data they have, but this approach sounds unsatisfactory to me. It has worked for this study because characterising the impact of gaseous absorbers was the focus and thus collecting long datasets while filtering was possible. In reality, people who have purchased the CAPS PMex want to be measuring aerosol extinction and thus it is undesirable to run on filter for extended periods as suggested. Indeed even if users did run with a 50% duty cycle, it still may not completely allow bias correction for reasons discussed above.

- It appears to me that there could have been scope for developing a more complex baseline correction scheme to try and overcome some of the above limitations. For example, it is shown that the red wavelength PMex units and not impacted by gaseous absorbers. Could correlations between red and green/blue wavelength units have been used to add extra constraint?

- More discussion is needed related to the realistic accuracy that applying the new corrections provides. Given the findings in the paper, how can a user quantify the accuracy of their CAPS PMex blue/green aerosol extinction measurements if they don't have a simultaneous measurement from a monitor that was run on filter?

- More discussion is needed on alternative approaches to enable reduction/elimination of biases, including those adopted by other users (e.g. scrubbing, gas reference channels). Rather than presenting a single solution, the paper would be enhanced by presenting a range of solutions that users could consider implementing to improve the quality of their data (with accompanied discussion on the merits/complications of each).

- Page 2, lines 9/10: The referenced work of Petzold et al., as far as I can ascertain, only undertook characterisation of a 630nm CAPS PMex unit. It would be worth stating this explicitly, particularly given that the biases in this work are only seen for units operating at wavelengths where the NO2 absorbs strongly.

- Page 2, line 13: here and throughout the manuscript the paper refers to the gaseous absorption leading to a baseline drift. I think this is confusing. The baseline in these

instruments is determined by such quantities as mirror cleanliness, mirror alignment, and near-constant Rayleigh scattering. Over time the baseline may drift due to temperature, ambient pressure changes etc. However, I view the NO2 absorption bias as more akin to a signal measurement. It doesn't drift, but rather is a quantity that varies dynamically and, as shown in the paper, with strong correlation to the particle signal of interest. I would prefer to have separation in terminology for these two distinct causes of error i.e. drift vs gaseous absorber signal.

- Section 3: can you explain why the ascending flank is steeper than the descending flank (line 19)?

- Section 3.2: do the data presented in Figure 3 represent an independent test of the correction scheme. i.e. are the data that were used to tune the scheme the same data that have been plotted in the histograms?

- Conclusions: I think the conclusions could be seen to provide contradictory guidance to users currently. On the one hand they suggest the new corrections could allow reduced frequency of baseline periods, but on the other suggest users should spend a lot of time filtering in order to characterise backgrounds adequately. I think the paper needs to more clearly describe that, in the absence of scrubbing of gaseous absorbers, users will never do better than having a designated CAPS measuring the gas phase background. If setups have less than this then it could come with cost in terms of residual errors from gaseous absorbers.

- Figure 4: the biases described in the bulk of this paper impact measurement accuracy rather than precision. I think the Allan Variance analysis in Figure 4 risks confusing readers with respect to understanding the absolute uncertainty of measurements. For example, it needs to be made clearer than despite the left hand panel of figure 4 suggesting a 1 sigma precision of around 0.1Mm-1, the total measurement uncertainty could be a lot bigger for these measurements.

---

## Referee Comment (RC2) · Anonymous Referee #1 · 8 Dec 2019

Review of SPfeifer et al. The influence of the baseline drift on the resulting extinction values of a CAPS PMex

This technical note reports the observed baseline drift of CAPS PMex instruments during a 2week sampling period at an urban location. The baseline drift at the shorter wavelengths are attributed to the varying amount of NO2 in the ambient air throughout the day. The authors proposed the use of cubic smoothing splines to interpolate and smooth the periodic background measurements and concluded that the new method help reduce the artifact caused by the varying baseline values thus provide more reliable extinction measurements.

In general, this manuscript lacks thorough discussion of the observations and detailed description of the methodology. In particular, I found the figures and their captions in poor quality and are not self-explanatory. The captions need to be substantially expanded to include more details of what are presented in the figures. Figure 2 misses the purple traces on the top panel, and is hard to interpret because the caption is inadequate.

Also, the proposed baseline correction method should be validated via simultaneous continuous baseline measurement before the authors can conclude that the corrected baseline represents the true values.

More descriptions of the cubic smoothing splines are needed. What are the chosen parameters for each instrument, and how were the data during the missing baseline period interpolated?

Figure 1: are the gradually increasing baseline of the Loss signal at 530 nm between 9/21 and 9/26 due to contamination building up on the cavity mirrors? Why is it only observable at the green wavelengths but not the other two? What causes the red loss signal to drift if it is not related to the NO2 level in the ambient air? These points should be thoroughly discussed in the text with great detail.

**Technical Corrections:**

Page 1 Line 6: where -> were

- Page 2 Line 2: which only -> which not only
- Page 3 Line17: The use of "carrier gas" is a bit confusing here. Change it to "ambient air".
- Page 4 Line 14: delete the duplicate "for the"

Page 4 Line 17: I don't see any "secondary plot" in Figure 4

---

## Short Comment (SC1) · 5 Jan 2020

This reply was posted incorrectly and refers to Anonymous Referee #1.

———————————————

---

## Author Comment (AC1) · 5 Jan 2020

We would like to thank the Referee for the constructive comments.

Please find our response to each of the comments below:

*The captions need to be substantially expanded to include more details of what are presented in the figures. Figure 2 misses the purple traces on the top panel, and is hard to interpret because the caption is inadequate.*

Yes, this is right in some ways. The purple traces are present but not really visible because they do not differ from the new calculated baseline using this x/y-scaling.

We have revised the plot and caption. The values for the measuring period are displayed subtracted by the Rayleigh scattering value. The scaling for the plot with lastbaseline is therefore better. The values for baseline periods (as predictor variables) are more visible, as well as any deviation between the interpolated baseline and values for the measurement period.

*Also, the proposed baseline correction method should be validated via simultaneous continuous baseline measurement before the authors can conclude that the corrected baseline represents the true values.*

That's exactly what we did. As stated out in the article, a particle filter was installed upstream of all three instruments. This is nothing else than a permanent baseline measurement. Two identical devices with the same wavelength were not available and are not necessary with this approach.

Any method can be considered as sufficient if it results in a (noisy) time series closed to zero without any artefacts, when using a particle filter upstream a particle extinction monitor.

Let me emphasize again at this point: Only the data points of the baseline measurments were used as the input for the cubic spline. The data points of the measuring period were used as a test case, as reference value, which should be reproduced by the new method. This should also be noticeable with the revised plot.

*More descriptions of the cubic smoothing splines are needed. What are the chosen parameters for each instrument, and how were the data during the missing baseline period interpolated?*

This is a legitimate point of criticism (see also Referee 2). Of course, we will not explain the mathematical concept of cubic smoothing splines. The function that is used is referenced in the text. But you are right. The parameters that are used should be specified in more detail. We also add a short section for the limitation of this approach (see Referee 2).

For the case with 1 Hz sampling rate, a baseline period of 5 min, and a duration of 1 minute, the smoothing parameters used were 1.1, 1.3, and 1.4 for the blue, green and red, respectively. These values were determined by minimizing the artefacts of a separate test dataset.

We have revised the specific section:
"A free smoothing parameter ($spar$) must be chosen, which depends on many factors, e.g. baseline period and duration but also on sampling rate and device noise etc.. Therefore, a suitable parameter must be found for each individual device and application. For the case with $1$ sampling rate, a baseline period of $5$ and a duration of $1$, the smoothing parameter used were 1.1, 1.3, and 1.4 for the blue, green and red, respectively. These values were determined by minimizing the artefacts of a separate test dataset. Alternatively it is also possible to determine a smoothing parameter automatically from the time series of baseline using for example the implemented generalized cross-validation method (GCV). The resulting values of the automatically calculated smoothing parameters using the GCV method do not differ significantly from the first method with values of 1.06 (blue), 1.25 (green) and 1.3 (red). Furthermore, all distinct data points with $1$ sampling rate were used (all.knots=TRUE). All other parameters were set to default. A complete description of the function can be found in the R Documentation (R Core Team, 2013)."

*Figure 1: are the gradually increasing baseline of the Loss signal at 530 nm between 9/21 and 9/26 due to contamination building up on the cavity mirrors? Why is it only*

*observable at the green wavelengths but not the other two? What causes the red loss signal to drift if it is not related to the NO2 level in the ambient air? These points should be thoroughly discussed in the text with great detail.*

Indeed, the increase in the second week is interesting. Because a particle filter was used it can be concluded that this is not due to contamination by aerosol particles on the cavity mirrors.

The structure for red also looks interesting. But it should be noted, however, that the signal here only fluctuates by 3 Mm-1 within a week. This is smaller by several orders of magnitude. However, no further data are available for a detailed analysis. It is also somewhat inappropriate, since the focus of the article is more on the effects of variability on extinction values, rather than their causes.

We have added in the article:
"The steady increase of the loss of CAPS-green in the second week is significant. The reason for this is unknown. Because a particle filter was used, it can be concluded that this is not due to contamination by aerosol particles on the cavity mirrors."

*Page 1 Line 6: where → were*

We have replaced "where" by "were".

*Page 2 Line 2: which only -> which not only*

We have inserted "not".

*Page 3 Line17: The use of "carrier gas" is a bit confusing here. Change it to "ambient air".*

We have replaced "carrier gas" by "ambient air".

*Page 4 Line 14: delete the duplicate "for the"*

We have deleted the duplicate "for the".

*Page 4 Line 17: I don't see any "secondary plot" in Figure 4*

We have deleted "secondary". We have also replaced "plot" by "results" in the next sentence.

---

## Author Comment (AC4) · 6 Jan 2020

Please find the revised plot of Fig. 2 as supplement.
* * *
[Figure]

[Figure]

**Fig. 1.**

---

## Referee Comment (RC3) · Anonymous Referee #3 · 3 Feb 2020

Review of "The influence of the baseline drift on the resulting extinction values of a CAPS PMex" by Pfeifer et al.

The manuscript describes ambient measurements of aerosol extinction (450, 530, and 630 nm wavelengths), black carbon mass, and $NO_2$ mixing ratio that were conducted in an urban environment over a two week period. As would be expected, the blue and green extinction measurements were most susceptible to variability in the $NO_2$ mixing ratios, while the red extinction measurement showed little sensitivity. It is well known that absorbing gases can change the CAPS measurement, which is why the instrument employs a simple background loss correction scheme. The authors find that this simple step-wise background correction does not keep up with the observed gas-phase variability, and so they employ a smooth spline to represent the background

loss over time. They report that this method reduces transients and artifacts in the extinction time series. The paper is very short, and the main finding seems to be that a cubic fit captures the timeseries variability of the CAPS background loss better than a 5-minute stepwise function. This is obvious and the sort of thing that I would expect to see as a 1-2 sentence statement in the Methods section of a journal paper, but not as a standalone paper (even one described as a technical note).

I do not think that the manuscript meets the journal's requirement for scientific significance – "Does the manuscript represent a substantial contribution to scientific progress within the scope of this journal (substantial new concepts, ideas, methods, or data)?" Consequently, I recommend that the paper be rejected.
* * *

---

## Author Comment (AC5) · 5 Feb 2020

We would like to thank the Referee for the comment.

We regret that the reviewer categorically rejects the analysis of measurement artifacts for the CAPSPMex for atmospheric measurements, as well as the presentation and description of a possible solution for a journal with focus on atmospheric measurement techniques.

The two points of criticism are somewhat contradictory, on the one hand the paper is said to be very short, on the other hand it is pointed out that the improvements are obvious, to be expected as an "1-2 sentence statement in the Methods section of a journal paper". This relativization is also in contradiction with the first two referees,

who would like to see a somewhat detailed description and/or a critical analysis of potential limitations of the correction scheme.

Indeed, the paper is truly a compact and concise presentation of a specific problem (intended as a technical note). The problem is analyzed and quantified. A possible solution is presented and sufficiently discussed. The resulting improvements are significant and cannot be ignored (as it seems also from the referee).

To the best of our knowledge, we do not know any scientific article that deals with the measurement artifact of a CAPSPMex for atmospheric conditions and discusses a user-friendly solution in post-processing. We consider the improvement of CAPSP-Mex data quality to be scientifically relevant and as a "substantial new (or alternative) method"

---

## Referee Comment (RC4) · Anonymous Referee #4 · 7 Feb 2020

This paper reports an alternative method to reduce the effect of baseline drift of light scattering measurement by the CAPS PMex monitors at different wavelength. This kind of study is very important for users of these instruments and contributes to the accurate monitoring of optical properties of aerosol particles. However, as suggested by previous reviewers, I believe that the novelty is insufficient as a full paper. More comprehensive analyses of the performance of these instruments, including accuracy of the instrument, will be necessary.
* * *

---

## Author Response (AR1)

February 28, 2020

We would like to thank the Referees for the constructive comments. Please find our response to each of the comments below:

Referee comments are denoted with RC. Author replies/comments are in blue and denoted with AC. Changes in the manuscript are in blue and italicized.

**1 Anonymous Referee #1**

RC: The captions need to be substantially expanded to include more details of what are presented in the figures. Figure 2 misses the purple traces on the top panel, and is hard to interpret because the caption is inadequate.
AC: Yes, this is right in some ways. The purple traces are present but not really visible because they do not differ from the new calculated baseline using this x/y-scaling.
We have revised the plot and caption. The values for the measuring period are displayed subtracted by the Rayleigh scattering value. The scaling for the plot with lastbaseline is therefore better. The values for baseline periods (as predictor variables) are more visible, as well as any deviation between the interpolated baseline and values for the measurement period.

RC: Also, the proposed baseline correction method should be validated via simultaneous continuous baseline measurement before the authors can conclude that the corrected baseline represents the true values.
AC: That's exactly what we did. As stated out in the article, a particle filter was installed upstream of all three instruments. This is nothing else than a permanent baseline measurement. Two identical devices with the same wavelength were not available and are not necessary with this approach.

Any method can be considered as sufficient if it results in a (noisy) time series closed to zero without any artefacts, when using a particle filter upstream a particle extinction monitor.
Only the data points of the baseline measurments were used as the input for the cubic spline. The data points of the measuring period were used as a test case, as reference value, which should be reproduced by the new method. This should also be noticeable with the revised plot.

RC: More descriptions of the cubic smoothing splines are needed. What are the chosen parameters for each instrument, and how were the data during the missing baseline period interpolated?
AC: This is a legitimate point of criticism (see also Referee 2). Of course, we will not explain the mathematical concept of cubic smoothing splines. The function that is used is referenced in the text. But you are right. The parameters that are used should be specified in more detail. We also add a short section for the limitation of this approach (see Referee 2).
For the case with 1 Hz sampling rate, a baseline period of 5 min, and a duration of 1 minute, the smoothing parameters used were 1.1, 1.3, and 1.4 for the blue, green and red, respectively. These values were determined by minimizing the artefacts of a separate test dataset.

We have revised the specific section:
*A free smoothing parameter (spar) must be chosen, which depends on many factors, e.g. baseline period and duration but also on sampling rate and device noise etc.. Therefore, a suitable parameter must be found for each individual device and application. For the case with $1\,\mathrm{Hz}$ sampling rate, a baseline period of $5\,\mathrm{min}$ and a duration of $1\,\mathrm{min}$, the smoothing parameter used were 1.1, 1.3, and 1.4 for the blue, green and red, respectively. These values were determined by minimizing the artefacts of a separate test dataset. Alternatively it is also possible to determine a smoothing parameter automatically from the time series of baseline using for example the implemented generalized cross-validation method (GCV). The resulting values of the automatically calculated smoothing parameters using the GCV method do not differ significantly from the first method with values of 1.06 (blue), 1.25 (green) and 1.3 (red). Furthermore, all distinct data points with $1\,\mathrm{Hz}$ sampling rate were used (all.knots=TRUE). All other parameters were set to default. A complete description of the function can be found in the R Documentation (R Core Team, 2013).*

RC: Figure 1: are the gradually increasing baseline of the Loss signal at 530 nm between 9/21 and 9/26 due to contamination building up on the cavity mirrors? Why is it only observable at the green wavelengths but not the other two? What causes the red loss signal to drift if it is not related to the NO2 level in the ambient air? These points should be thoroughly discussed in the text with great detail.
AC: Indeed, the increase in the second week is interesting. Because a particle filter was used it can be concluded that this is not due to contamination by aerosol particles on

the cavity mirrors.

The structure for red also looks interesting. But it should be noted, however, that the signal here only fluctuates by 3 Mm-1 within a week. This is smaller by several orders of magnitude. However, no further data are available for a detailed analysis. It is also somewhat inappropriate, since the focus of the article is more on the effects of variability on extinction values, rather than their causes.

We have added in the article:
*The steady increase of the loss of CAPS-green in the second week is significant. The reason for this is unknown. Because a particle filter was used, it can be concluded that this is not due to contamination by aerosol particles on the cavity mirrors.*

RC: Page 1 Line 6: where → were
AC: We have replaced *where* by *were.*

RC: Page 2 Line 2: which only → which not only
AC: We have inserted *not.*

RC: Page 3 Line17: The use of "carrier gas" is a bit confusing here. Change it to "ambient air".
AC: We have replaced *carrier gas* by *ambient air.*

RC: Page 4 Line 14: delete the duplicate "for the"
AC: We have deleted the duplicate *for the.*

RC: Page 4 Line 17: I don't see any "secondary plot" in Figure 4
AC: We have deleted "secondary". We have also replaced "plot" by "results" in the next sentence.

**2 Anonymous Referee #2**

Response to the general comments:
The referee has two relevant critical points. First, a more detailed description of other alternatives, especially hardware based solutions. Furthermore, any limitations of the correction method are not sufficiently described. Both points are generally legitimate remarks.

It is undisputed that a functioning hardware based solution is preferable to a software-based correction. However, it should be emphasized that this correction method is a simple method without any additional cost and effort to optimize the measured values. In addition, it is even possible to optimize old existing data sets in a post-processing. For the reasons given above, we consider this method to be useful. The aim of this article is not to gain "ingenious novel insights". The focus of this article is to describe

the effect of the baseline drift on the resulting extinction values, exemplary for an urban background station. The intention is to analyze the artefacts and to present a simple method to reduce these effects. This is in the sense of a technical note. We have tried to consider the points of criticism, without expanding or shifting the focus of the article.

RC: The suggested correction scheme works for the example data provided. However it will not work as effectively when there is significant variability in NO2 between consecutive baseline measurements. Indeed, it would appear possible that in some circumstances the applied correction could make the bias worse compared to the standard method (e.g. where the baseline increases between consecutive baseline periods but gaseous absorption decrease in the interim period). For this reason I question what applying this new correction really enables users to say with confidence about the accuracy of their resulting extinction numbers. This needs to be examined in more detail in the paper.

AC: In general, this fact is correct. The authors are aware that any form of interpolation never leads to a gain in information. This is a fundamental fact and should be clear to everyone who uses interpolation no matter where it is used. However, if the variability between two baseline periods fluctuates strongly, the internal calculation also fails. The internal calculation is also based on the assumption of a constant baseline for the following measuring period. In the broadest sense this is also just an interpolation, more precisely it is a forward extrapolation. It is also correct that under extreme conditions: strongly fluctuating predictor variable with unfavourable choice of the smoothing parameter the interpolation procedure can lead to ringing/overshot structure. However, it is important that in any case the period for the baselines must resolve the variability (this fact should be clear to the user and is even mentioned in the manual). The new procedure only allows to consider the drift or trend between two baselines. If, however, this is guaranteed, the interpolation delivers better results than the internal calculation.

To make this fact clear to the reader, the manuscript has been changed as follows: *It should be emphasized that the use of splines interpolation to recalculate the baseline has its limits. Only trends that can be estimated from the baseline data can be reproduced for lastbaseline. It is impossible to reproduce any faster fluctuations that are not covered by the selected baseline period and duration. Furthermore, there is the possibility that under extreme conditions with strongly fluctuating baseline trends the method can lead to overshot structures. In these cases, the first step should be the readjustment the baseline settings.*

RC: To overcome the above, the authors suggest that users will need to tune their smoothing parameters based on the data they have, but this approach sounds unsatisfactory to me. It has worked for this study because characterising the impact of gaseous absorbers was the focus and thus collecting long datasets while filtering was possible. In reality, people who have purchased the CAPS PMex want to be measuring aerosol extinction and thus it is undesirable to run on filter for extended periods as suggested. Indeed even if users did run with a 50% duty cycle, it still may not completely allow bias correction for reasons discussed above.

AC: There is probably a misunderstanding here.
The filtering upstream was only done in the context of this experiment. First, to completely exclude the influence of aerosol particles on any effect. Second, to generate a reference for testing. The device should give (more or less symmetrically distributed noisy) values around zero, and no artefacts. No particle filter is required for normal operation. Just the existing data of zero period are used as predictor variables for the cubic smoothing spline. This means no additional losses of data points.
According to the comment of second referee the Fig. 2 was revised. One can see very well how the spline even reproduce the trend during a baseline measurement. We ve added this point also in text.

The corresponding section has been revised (also according to Referee 1):
*A free smoothing parameter (spar) must be chosen, which depends on many factors, e.g. baseline period and duration but also on sampling rate and device noise etc.. Therefore, a suitable parameter must be found for each individual device and application. For the case with $1\,\mathrm{Hz}$ sampling rate, a baseline period of $5\,\mathrm{min}$ and a duration of $1\,\mathrm{min}$, the smoothing parameter used were 1.1, 1.3, and 1.4 for the blue, green and red, respectively. These values were determined by minimizing the artefacts of a separate test dataset. Alternatively it is also possible to determine a smoothing parameter automatically from the time series of baseline using for example the implemented generalized cross-validation method (GCV). The resulting values of the automatically calculated smoothing parameters using the GCV method do not differ significantly from the first method with values of 1.06 (blue), 1.25 (green) and 1.3 (red). Furthermore, all distinct data points with $1\,\mathrm{Hz}$ sampling rate were used (all.knots=TRUE). All other parameters were set to default. A complete description of the function can be found in the R Documentation (R Core Team, 2013).*

RC: It appears to me that there could have been scope for developing a more complex baseline correction scheme to try and overcome some of the above limitations. For example, it is shown that the red wavelength PMex units and not impacted by gaseous absorbers. Could correlations between red and green/blue wavelength units have been used to add extra constraint?
AC: The deeper sense of such correlation is not clear to us. But as already mentioned in a point above, we refer to an alternative method to determine a suitable smoothing parameter using generalized cross-validation method (GCV).

RC: More discussion is needed related to the realistic accuracy that applying the new corrections provides. Given the findings in the paper, how can a user quantify the accuracy of their CAPS PMex blue/green aerosol extinction measurements if they don't have a simultaneous measurement from a monitor that was run on filter?
AC: The user can estimate the deviations by comparing the recalculated with the internal baseline values. Because the baseline correction is additive, the difference between both baseline values is the absolute deviation of the resulting extinction value. This is already explained in the text.

RC: More discussion is needed on alternative approaches to enable reduction/elimination of biases, including those adopted by other users (e.g. scrubbing, gas reference channels). Rather than presenting a single solution, the paper would be enhanced by presenting a range of solutions that users could consider implementing to improve the quality of their data (with accompanied discussion on the merits/complications of each).

AC: To the best of our knowledge we don't know any no scientific publication dealing with scrubbing of NOx in a CAPS PMex.

It would probably also be accompanied by a modification of the aerosol, in particular an increase of particle losses. A combination of gas monitor, in particular a device of identical construction, a CAPS - NO2 Monitor, in combination with CAPS PMex seems to be an alternative. DeFaria et al. (2017) combined two instruments but with different wavelength, a CAPS PMex with 630nm. To the best of our knowledge we don't know any further scientific publication dealing with this combination. Despite both devices, the periodic baseline is still required due to the different influencing factors (mentioned by the referee) in both devices, in particular the contamination of the mirror by aerosol particles in the CAPS PMex. The gas monitor can best be used as a reference, e.g. to point out any strong fluctuations of absorbing gases. However, this means that the consideration presented in this article is not obsolete but still relevant.

Due to a lack of literature and the reason mentioned above, we have only slightly changed the article:

*The use of a gas monitor in parallel operation can serve as a reference to adjust the baseline period. However, the new method can be used to take into account the continuous change of the background signal and improve the quality of the resulting extinction values.*

RC: Page 2, lines 9/10: The referenced work of Petzold et al., as far as I can ascertain, only undertook characterisation of a 630nm CAPS PMex unit. It would be worth stating this explicitly, particularly given that the biases in this work are only seen for units operating at wavelengths where the NO2 absorbs strongly.

AC: We consider the specification of the wavelength of each single cited publication with a CAPS PMex in the context of the introduction as unnecessary. In addition, it is already pointed out several times in the text that the larger deviations occur for 450 and 520nm.

RC: Page 2, line 13: here and throughout the manuscript the paper refers to the gaseous absorption leading to a baseline drift. I think this is confusing. The baseline in these instruments is determined by such quantities as mirror cleanliness, mirror alignment, and near-constant Rayleigh scattering. Over time the baseline may drift due to temperature, ambient pressure changes etc. However, I view the NO2 absorption bias as more akin to a signal measurement. It doesn't drift, but rather is a quantity that varies dynamically and, as shown in the paper, with strong correlation to the particle signal of interest. I would prefer to have separation in terminology for these two distinct causes of error i.e.

drift vs gaseous absorber signal.

AC: It's a question of point of view. If deviations from near-constant Rayleigh scattering and/or temperature and pressure influence are attributed to the drift, why not the influence of absorbing gases? The focus of this article are particle extinction monitor (extinction by aerosol particles). Therefore we keep the terminology, although we understand the intentions of the referee.

RC: Section 3: can you explain why the ascending flank is steeper than the descending flank (line 19)?

AC: A possible explanation would be: Traffic peaks as sources arise quickly and directly. Dilution process as a sink is a relative slow process. However, no further data are available for interpretation.

RC: Section 3.2: do the data presented in Figure 3 represent an independent test of the correction scheme. i.e. are the data that were used to tune the scheme the same data that have been plotted in the histograms?

AC: No, maybe it's badly phrased. Just the data points from baseline periods are used as predictor variables. The data points from measurement periods are used for testing. The smoothing parameter are determined by a separate test data set (see previous point above).

RC: Conclusions: I think the conclusions could be seen to provide contradictory guidance to users currently. On the one hand they suggest the new corrections could allow reduced frequency of baseline periods, but on the other suggest users should spend a lot of time filtering in order to characterise backgrounds adequately. I think the paper needs to more clearly describe that, in the absence of scrubbing of gaseous absorbers, users will never do better than having a designated CAPS measuring the gas phase background. If setups have less than this then it could come with cost in terms of residual errors from gaseous absorbers.

AC: Indeed, the last section is really badly phrased and therefore contradictory. However, at this point the consequences resulting from the interpolation should be presented. The supposedly contradictory statement is a result of the qualitative gain and the limitation of the interpolation

Undisputed, for low variable background conditions (variability is significantly smaller than the baseline period) this approach enable the possibility to reduce the baseline periods. On the other hand, the background variability in the majority of cases is unknown and partially coupled. So, if one considers the background signal as equivalent, the measuring and baseline periods should be equally weighted.

The last section has been rewritten to make it more precise:

*If the change of the background signal is relatively slow, the new method allows to reduce the frequency of baseline periods and thus reduce number of position changes of the built-in ball valve extending its lifetime. On the other hand, in the majority of cases, the background variability and the ambient aerosol and the composition of the carrier gases*

*may be closely coupled (e.g. near traffic emission). From this it follows that the measuring and baseline period should be equally weighted, if one considers the background signal as equivalent. The use of a gas monitor in parallel operation can serve as a reference to adjust the baseline period. However, the new method can be used to take into account the continuous change of the background signal and improve the quality of the resulting extinction values.*

RC: Figure 4: the biases described in the bulk of this paper impact measurement accuracy rather than precision. I think the Allan Variance analysis in Figure 4 risks confusing readers with respect to understanding the absolute uncertainty of measurements. For example, it needs to be made clearer than despite the left hand panel of figure 4 suggesting a 1 sigma precision of around 0.1Mm-1, the total measurement uncertainty could be a lot bigger for these measurements.

AC: Also at this point, it seems to be a question of point of view. Of cause, for a single measurement period the effect should be attributed to the accuracy than precision. On a larger time scale, this effect averages out (due to the pos. and neg. deviation, depending on increasing or decreasing baseline or loss values). This just results in a spread of values, which should be attributed to precision. The text explicitly distinguishes between the averaged results and the maximum deviations (accuracy) achieved.

**3 Anonymous Referee #3**

RC: The paper is very short, and the main finding seems to be that a cubic fit captures the timeseries variability of the CAPS background loss better than a 5-minute stepwise function. This is obvious and the sort of thing that I would expect to see as a 1-2 sentence statement in the Methods section of a journal paper, but not as a standalone paper (even one described as a technical note). I do not think that the manuscript meets the journal's requirement for scientific significance - "Does the manuscript represent a substantial contribution to scientific progress within the scope of this journal (substantial new concepts, ideas, methods, or data)?" Consequently, I recommend that the paper be rejected.

AC: We regret that the reviewer categorically rejects the analysis of measurement artifacts for the CAPSPMex for atmospheric measurements, as well as the presentation and description of a possible solution for a journal with emphasis on atmospheric measurement techniques.
The two points of criticism are somewhat contradictory, on the one hand the paper is said to be very short, on the other hand it is pointed out that the improvements are obvious, to be expected as an "1-2 sentence statement in the Methods section of a journal paper". This relativization is also in contradiction with the first two referees, who would like to see a somewhat detailed description and/or a critical analysis of potential limitations of the correction scheme.
Indeed, the paper is truly a compact and concise presentation of a specific problem (intended as a technical note). The problem is analyzed and quantified. A possible solution is presented and sufficiently analyzed. The resulting improvements are significant and cannot be ignored (as it seems also from the referee).

To the best of our knowledge, twe do not know any scientific article that deals with the measurement artifact of a CAPSPMex for for atmospheric conditions and discusses a user-friendly solution in post-processing. We consider the improvement of CAPSP-Mex data quality to be scientifically relevant and as a substantial new (or alternative) method.

**4 Anonymous Referee #4**

RC: This paper reports an alternative method to reduce the effect of baseline drift of light scattering measurement by the CAPS PMex monitors at different wavelength. This kind of study is very important for users of these instruments and contributes to the accurate monitoring of optical properties of aerosol particles. However, as suggested by previous reviewers, I believe that the novelty is insufficient as a full paper. More comprehensive analyses of the performance of these instruments, including accuracy of the instrument, will be necessary. AC: We are pleased to see that the Referee agrees on the usefulness and the great benefits of this alternative method.

Of course the novelty is limited. It is an alternative post-processing for an existing device using an established mathematical method, so the article was designed as a technical note from the beginning. There is no need for a totally new mathematical concept. This is also a great advantage, which allows the user to use a variety of already existing functions and libraries depending on the preferred programming language.

Regarding the comprehensive analysis of performance: The artifacts are primarily due to the fluctuating background in combination with the internal calculation procedure and less to other aspects of performance. We do not believe that a comprehensive performance analysis will improve the novelty of the paper. It would only blur the focus of the work.

Nevertheless, it may be useful to address aspects of quality assurance to assess performance and accuracy.

We have added two more sections and one more plot.
in "Experimental set-up":
"Before and after measurements the quality of the CAPS PMex were checked by a comparison with a thoroughly and regularly calibrated reference nephelometer (Ecotech Aurora 4000). For this purpose non-absorbing ammonium sulfate particles were used. The truncation error in the nephelometer has been corrected using the method of Müller et al. (2011). Nevertheless relatively small particles were generated (mean size of approx. 50 nm) to minimize the effect of truncation. Analogous to the comparison of the measured and mie calculated theoretical values using mono-disperse particles and a reference CPC (Petzold et al., 2013), correction factors can be derived by comparing the

*truncation-corrected scatter values of the reference nephelometer with the respective measured extinction values. The factors represent a correction of the internal calibration, which primarily considers the influence of the purge air variability.*

in "Results":
*"Before and after the measured time series the comparison of CAPS PMex and reference nephelometer show a small but very stable deviation, exemplary shown in Fig. 2. The devices show slightly too high values in the range of 3–4 %, 6–8 %, and 6–7 % for the blue, green and red, respectively.*

Apart from that, for a statement of the expected accuracy we think that the mentioned maximum values or percentiles of artifacts are sufficient. Concerning the precision (for long average periods) we consider the simple statistical data of mean value, standard deviation and skewness as well as Std.Allan-Var analysis as sufficient.

---

## Author Response (AR2)

April 1, 2020

We would like to thank the Referees for the constructive comments. Please find our response to each of the comments below:

Referee comments are denoted with RC. Author replies/comments are in blue and denoted with AC. Changes in the manuscript are in blue and italicized.

**1 Anonymous Referee #1**

RC: Figure 1: I suggest revising this plot to show extinction readings at three wavelengths in one panel, under the same y-axis scale, so that it's clearer to the readers the magnitude of the baseline variation at these different wavelengths. It will help determine whether the shift in CAPS-green in the second week is "significant", as stated in Page 3 Line 24. It is complicated to implement this, because all three devices have different basic loss signals (blue 580, green 380, red 480). With a second or third y-axis this could be manageable. However, we think that this is even more difficult for the reader to understand. Alternatively you can leave the multi-plot as it is and only adjust the scaling. but this results in an almost horizontal line for red. Not very informative.
So we leave the plot as it is.

RC: Page 3 Line 23: does the difference between CAPS-blue and CAPS-green extinction readings scale with the difference in NO2 absorption cross-sections at 450 and 530 nm? That analysis can help determine the source of the variation in the extinction signals. Unfortunately, no new facts can be derived from the existing database. Only that the increase is not due to NO2 or particle contamination of the mirrors.

RC: Page 4 Line 1: To be precise, the variations in the baseline is not caused by "changes in the gas composition", but rather the rapid change in the abundance of the absorbing gases, such as NO2, in the ambient air.

Perhaps the expression "change of gas composition" sounds very wide-ranging and exaggerated. We have therefore reformulated the term: *rapid change in the concentration of absorbing gases.*

RC: Figure 3 Caption: the caption does not mention that the results are from CAPS-blue.

we have specified the caption:

*Time series of lastbaseline (top) and the resulting extinction values (bottom) for the uncorrected (blue) and corrected method (red) for CAPS-blue (450 nm) measuring particle free ambient air.*

**2 Anonymous Referee #4**

RC: For the results of the comparison with the nephelometer, the authors described that "The factors represent a correction of internal calibration, which primarily consider the influence of the purge air variability.". This sentence seems unclear. Do you mean the flow rates of purge air or their mixing situations with sample air were different for the three different CAPS PMex?

We mean variability of the flow rate, to be more precise the ratio of the flow rates of purge air and sample air.

we have specified this in the text:

*The factors represent a correction of internal calibration, which primarily consider the influence of the variable ratio of purge air and sample air flow rate.*

RC: In addition, information on the calibrations of nephelometers and the corrections of the nephelometer data to adjust to those at the wavelengths of CAPS PMex should be added.

The nephelometer is calibrated with particle free air and $CO_2$. The wavelengths were interpolated using the scattering Angström exponent, although this is not necessary for blue 450nm. For the other two wavelengths the deviation is very small with 5nm (nephelometer: 525nm or 635nm).

We have added:

*using $CO_2$ as high span gas*

and:

*The values were adjusted adjusted to the wavelength of the respective CAPS PMex (450, 530 and 630 nm) by using scattering Angström exponent.*

RC: Also, the maximum values of extinction coefficients at wavelengths 450, 530, and

630 nm look similar. The results may be strange, because relatively large extinction Angstrom exponents are expected for small particles with diameters of about 50 nm.

The referee is absolutely right. The extinction Angstrom exponents are relatively large. As an explanation: For the devices, the manufacturer states that they have a linear behavior for values smaller than $500\,\mathrm{Mm}^{-1}$. Therefore, only values smaller than $500\,\mathrm{Mm}^{-1}$ were used for the quality checks, to ensure comparability of the results of all the devices. We have added:

*In order to reduce the influence of potential non-linear effects of CAPS PMex, only values less than $500\,\mathrm{Mm}^{-1}$ were used for this analysis.*

RC: For baseline corrections of this kind of instruments, the linear interpolation method is widely (normally) used. The comparison of results of the linear interpolation method and cubic spline method should be added.

As already pointed out in the text, there are many methods of interpolation, with all their advantages and disadvantages. Linear interpolation is one of them In contrast to various other requests, this request is within the scope of this work and would not shift the focus of the work. Nevertheless, this requires a major change in many sections of the text as well as some plots.

In Fig. 3 und 4 the results for the linear interpolation were added. The graphics have been slightly changed and the captions have been modified.

In the abstract we changed the last section:
*Two methods are shown to minimize this effect. Modified continuous baseline values are calculated in a post-processing step using simple linear interpolation and cubic smoothing splines. Both methods are useful to reduce artefacts, although the use of cubic smoothing splines gives slightly better results. The extinction artefacts are diminished and the effective scattering of the resulting extinction values is reduced by about $50\,\%$.*

In chapter 3.2, P. 4 L. 18:
*It is possible to reduce these artefacts by using interpolation methods. Two different procedures were considered. The first one is a simple and often used linear interpolation method. A potentially better alternative is the use of cubic smoothing splines.*

In chapter 3.2, P. 4 L. 32:
*It should be emphasized that the use of any interpolation method to recalculate the baseline has its limits. Only trends that can be estimated from the baseline data can be reproduced for lastbaseline. It is impossible to reproduce any faster fluctuations that are not covered by the selected baseline period and duration. Furthermore, when using the cubic smoothing splines there is the possibility that under extreme conditions with strongly fluctuating baseline trends the method can lead to erroneous overshot structures. In these cases, the first step should be the readjustment the baseline settings.*
*If the requirements are fulfilled, these approaches result in a continuous time series of current baseline values, without phase-shift relative to the loss signal (see Fig. ??). As expected, the result is slightly better when cubic smoothing splines are used, as this is*

*a continuously differentiable function. Another important difference is that with cubic smoothing splines trends during a baseline measurement are considered and are therefore reproducible. In contrast to this, the linear interpolation method uses only one average value per baseline measurement, analogous to the internal procedure. As a result, there are individual cases where the linear interpolation does not lead to any improvement of the extinction values, but there are also cases where the improvement corresponds to that of the cubic smoothing spline. However, in both case the resulting extinction values improve significantly. In Fig. ?? the resulting histograms and statistical parameters for particle extinction for all instruments and the entire time series are shown. As expected, the mean value remains almost unchanged at values close to zero. But the distribution becomes narrower and more symmetrical. For the CAPS-blue the standard deviation is reduced by 43 % using the linear interpolation and 50 % using cubic smoothing splines. For CAPS-green the reduction is 19 % with both methods. The skewness for CAPS-blue is reduced from 2.909 to a value of 0.756 using the linear interpolation and 0.104 for cubic smoothing splines. The results for the CAPS-red remain almost unchanged.*

In chapter 3.2, P. 5 L. 24:
*For instance, at 450 nm, the precision is improved by a factor of approximately 2.8 using the linear interpolation method and 3.7 using cubic smoothing splines.*

In chapter 4, P. 5 L. 31:
*The use of linear interpolation or cubic smoothing splines to calculate the current baseline values are more adequate methods for a variable background. Both procedures lead to improved values, although the result for cubic smoothing splines is slightly better. Artefacts for particle extinction almost disappear and variability decreases.*

In chapter 4, P. 5 L. 34:
*If the change of the background signal is relatively slow, these methods allows to reduce the frequency of baseline periods and thus reduce number of position changes of the built-in ball valve extending its lifetime. On the other hand, in the majority of cases, the background variability is unknown and the ambient aerosol and the composition of the carrier gases may be closely coupled (e.g. near traffic emission). From this it follows that the measuring and baseline period should be equally weighted, if one considers the background signal as equivalent. The use of a gas monitor in parallel operation can serve as a reference to adjust the baseline period. However, these interpolation methods, in particular cubic smoothing splines, can be used to take into account the continuous change of the background signal and improve the quality of the resulting extinction values.*